# Mechanisms of Rhinovirus Neutralisation by Antibodies

**DOI:** 10.3390/v13030360

**Published:** 2021-02-25

**Authors:** Lila Touabi, Faryal Aflatouni, Gary R. McLean

**Affiliations:** 1Cellular and Molecular Immunology Research Centre, London Metropolitan University, London N7 8DB, UK; lit0208@my.londonmet.ac.uk (L.T.); faa1280@my.londonmet.ac.uk (F.A.); 2National Heart and Lung Institute, Imperial College London, London W2 1PG, UK

**Keywords:** rhinovirus, antibodies, neutralisation, vaccine

## Abstract

Antibodies are a critical immune correlate of protection for rhinoviruses, particularly those antibodies found in the secretory compartment. For nonenveloped viruses such as rhinoviruses, antibody binding to regions of the icosahedral capsid can neutralise infections by a number of different mechanisms. The purpose of this review is to address the neutralising mechanisms of antibodies to rhinoviruses that would help progress vaccine development. At least five mechanisms of antibody neutralisation have been identified which depend to some extent on the antibody binding footprints upon the capsid. The most studied mechanisms are virion aggregation, inhibition of attachment to cells, and stabilisation or destabilisation of the capsid structure. Newer mechanisms of degradation inside the cell through cytoplasmic antibody detection or outside by phagocytosis rely on what might have been previously considered as non-neutralising antibodies. We discuss these various approaches of antibody interference of rhinoviruses and offer suggestions as to how these could influence vaccine design.

## 1. Introduction

Picornaviruses are a group of small single-stranded positive-sense nonenveloped RNA viruses that infect vertebrates including birds and mammals. These related viruses make up the family *Picornaviridae* that contains 147 species that are grouped into 63 genera (www.picornaviridae.com). It is one of the largest virus families containing notable human viruses such as polioviruses, coxsackieviruses, enteroviruses, hepatitis A viruses (HAV), and rhinoviruses (RVs). A notable animal virus, foot-and-mouth disease virus (FMDV) is also part of the family. All viruses have a conserved genome organisation of one large open reading frame (ORF) ranging in size between 6.7 kb and 10.1 kb, a similar replication life cycle, and homologous icosahedral capsid structures consisting of 60 protomers of four polypeptides (VP1, VP2, VP3, and VP4).

## 2. Rhinoviruses

The *Enterovirus* genus of *Picornaviridae* contains 15 virus species, including the three human rhinovirus (RV) species known as types A, B and C (RV-A, RV-B, RV-C) that are classified genetically. Within these groupings, there are approximately 160 serotypes or strains of RV that can also be classified by entry receptor use in addition to structural proteins genome sequence (Table 1). These viruses cause more than half of upper respiratory tract infections, known as the common cold, and have a huge economic and social impact on society through unnecessary medical treatments and missed days from school and work [1]. Common colds due to RV infection were estimated to have caused 20 million days of missed work, 22 million days of missed school, and 27 million physician visits in the United States in 1996 alone [2]. RV infection occurs year-round, amongst all age groups and can be transmitted by direct contact, through aerosol particles or via contaminated objects. However, RV infections are most dominant between the northern hemisphere autumn (September and October) and spring (March and April). RVs mostly cause upper respiratory tract illnesses with symptoms including sore throat, rhinorrhoea, sneezing, cough, nasal congestion, and headache [2]. RV has also been associated with lower respiratory tract infections causing pneumonia, bronchiolitis and can cause breathing difficulties in individuals with respiratory conditions such as cystic fibrosis, asthma and chronic obstructive pulmonary disease (COPD). Thus, RV infections are important mediators of acute exacerbation of these chronic lung diseases (asthma and COPD) and can therefore be life-threatening [3,4]. Evidence indicates that the severity of the RV infection could depend on the species and serotype [5]. Some studies suggested that RV-C are associated with more severe respiratory illness in children when compared to RV-A and RV-B. Nevertheless, RV can be detected in both symptomatic and asymptomatic individuals [6].

RVs were first discovered in 1956 and named common cold viruses due to the upper respiratory tract symptoms caused after infection of humans [7]. Despite extensive research, specific antivirals have not been identified, therefore no targeted intervention exists for RV infections and the treatment options available are palliative [8]. Development of a RV vaccine has suffered the same fate and no protective vaccine has been approved after numerous attempts [9]. This is mostly due to the emergence of more than 160 serotypes and the extreme challenges associated with generation of cross-serotype immunity due to the high antigenic variability within the RV serotypes. Several vaccine clinical trials with inactivated viral preparations have failed to generate cross-serotype protective immunity and therefore have not been performed since the 1970s, however, newer experimental approaches have shown promising efficacy in preclinical studies [10]. These more recent approaches, involving subunit or inactivated vaccines containing up to 50 RV serotypes, have displayed efficacy in preclinical animal models, including RV challenge models [9].

## 3. Capsid Protomer Structure, Receptor Binding and Neutralising Ab Sites

The RV capsid, the target of protective Abs, is an icosahedral structure approximately 30 nm in diameter and is built from 60 copies of the viral proteins protomer unit consisting of VP1, VP2, VP3, and VP4 subunits. These are the only RV structural proteins with VP1, VP2 and VP3 being surface exposed and therefore the main targets for neutralising Ab responses, whereas VP4 lies underneath at the RNA/capsid interface [11]. For RV-A and RV-B the capsid surface contains several topological features, including a raised area known as the 5-fold symmetry axis and a 20Å deep depression known as the canyon, with its rims created by VP1 residues to the north and VP2/VP3 residues to the south. The canyon regions of the capsid contain the binding site for the cellular receptors for major group RV, intracellular adhesion molecule (ICAM-1) [12]. The canyon surrounds the 12 icosahedral 5-fold vertices formed by adjacent VP1 subunits and these are the binding sites for the minor group RV entry receptor low-density lipoprotein receptor (LDLR) [13]. RV-C contains several different capsid features with the most notable being a smaller 5-fold axis platform, a minimal canyon depression and a characteristic finger-like projection made up of VP1 and VP2 residues [14]. The entry receptor for RV-C has been identified as cadherin-related protein 3 (CDHR3) [15]. Studies have identified four neutralising immunogenic (NIm) sites as targets for Abs based on RV-B14 escape mutants, these sites are known as NIm-IA, NIm-IB, NIm-II, and NIm-III and map to surface regions of VP1, VP2 and VP3 [16,17]. Further experimentation extended the studies of RV-B14 NIms to RV-A2 that suggest similarly that NIm-1A and NIm-1B are part of VP1 and are positioned north of the canyon, that NIm-II is part of the VP2 PUFF loop and NIm-III is within VP3 close to the 2-fold axis [18,19]. Both NIm-II and NIm-III are located to the south of the canyon. The NIms have not been confirmed on any other RV serotypes as yet, including the RV type C strains, but are likely to be analogous at least for type A and B RVs although the amino acid sequences will likely vary considerably. Perhaps not surprisingly the NIms resemble those of related picornaviruses, HAV and poliovirus [20] and FMDV [11,21], which have similar capsid structures. See Figure 1 for a structure image and schematic of the RV-A16 protomer showing the various topological features, potential NIm sites and the entry receptor locations.

## 4. Mechanisms of RV Neutralisation by Abs

Neutralising antibodies (nAbs) either monoclonal or polyclonal are very important in the defence against viruses [22]. Frequently, nAbs prevent viral infection by binding directly to specific portions of viral capsid or surface projections and ultimately function to interfere with attachment and entry into host cells [23]. However only a small subclass of antibodies that bind to viruses are able to induce neutralisation and then only under certain conditions do nAbs function effectively. Furthermore, after viral infection the host takes approximately 1–2 weeks to produce highly effective nAbs but these do remain protective against any future infection with the same agent [24]. For RV, the speed of infection and transmission renders many of the Abs ineffective until re-exposure with the same serotype. RV-specific nAbs are notoriously serotype-specific due to the prevalence of antigenically distinct capsid sequences and the fact that Abs recognise highly variable surface exposed structures [9]. Such variability of capsid sequences has made RV vaccine development extremely challenging. Furthermore, to date no descriptions of neutralising Abs for RV-C have been described so we focus here on knowledge obtained for RV-A and RV-B types.

There are several different potential mechanisms of RV neutralisation by Abs and these will depend on the site of interaction with the capsid as well as the valency of interaction. Neutralising Abs against RV-B14 had previously been classed into three groups named as strong, intermediate and weak, based on their neutralising properties [25]. The strong nAbs appear to form stable monomeric virus/Ab complexes and bound bivalently to the virion surface via NIm-IA. Abs that precipitated virions were classified as intermediate neutralisers and weak neutralisers were shown to form unstable monovalent complexes with the virus, binding with a stoichiometry of 60 Abs per virion as opposed to the 30 Abs observed for strong nAbs. Monovalent interaction of a bivalent IgG could therefore potentially aggregate a number of viral particles, particularly over a certain range of Ab/virion ratio and would likely be more prevalent with polyclonal Ab preparations, however aggregation of RV [26] and the structurally related poliovirus [27] by monoclonal Abs has been shown. Depending on the epitope bound, nAbs could inhibit the attachment of viruses to host cells and thereby to their entry receptors through simple steric hindrance or by inducing capsid structural changes that abrogate interactions [28]. Binding of Abs has also been shown to cause the RV capsid conformational changes that trigger release of RNA prematurely before interaction with cells [29] or for poliovirus by inhibiting the RNA release into cells via capsid stabilisation, effectively preventing uncoating [30]. Another mechanism of Ab neutralisation of RV is through the antiviral protein tripartite motif containing-21 (TRIM21), a ubiquitin ligase and Ab receptor that is found inside cells [31]. TRIM21 is able to bind to the Fc region of antibodies attached to viruses and carried into cells, which in turn induces proteasomal degradation of the virions by generation of polyubiquitin chains [32]. Recently another mechanism of RV neutralisation has been revealed which involves the Fc receptor-dependent function of antibody-dependent cellular phagocytosis (ADCP) [33]. This particular mechanism is thought to contribute to the clearance of cells infected with viruses and might also facilitate antigen presentation to stimulate immune responses to RV infection [34].

Therefore, RV neutralisation by Abs has many aspects to consider. Not least, how the Abs bind to their epitopes. Does binding induce conformational changes to the capsid proteins? Do Abs inactivate the virus by blocking access and entry to cells irreversibly? Are Abs most affective against viruses in suspension or after entry into host cell? Below we describe the various mechanisms of RV neutralisation by Abs that have been discovered to date. For a summary, see Figure 2 that shows schematically these varied mechanisms of action of RV nAbs.

### 4.1. Abs That Block Attachment of RV to Cells

As outlined earlier, the RV capsid is the target of nAbs that bind to surface exposed regions of VP1, VP2 and VP3. The capsid protein VP1 has emerged as an important region as it contains two NIms and the sites for attachment to the two best characterised entry receptors, ICAM-1 and LDLR. Abs binding VP1, therefore, could block viral attachment to the host cell by interference. Interestingly, Abs against VP1 have successfully been shown to cross-react with numerous RV serotypes [35], increasing their potential for cross-serotype neutralisation [36].

The most commonly studied mechanism is hindrance of the RV-B14 capsid’s interaction with the cellular entry receptor ICAM-1 and of the RV-A2 capsid with the LDLR entry receptor. Structural studies have shown that Abs tend to bind bivalently across the 2-fold axis of symmetry that rigidifies or stabilises the capsid structure interfering with attachment. Thus, two mechanisms have been proposed where nAbs can inhibit binding of RV to entry receptors via nAb-induced stabilisation of the viral capsid [37,38] or by physical interference by the nAb with the entry receptor [28]. Structures solved with IgG [37] and Fab fragments [38] of the same Ab demonstrate the bivalent binding across the 2-fold axis and the potential for multiple positions of the Fc portion. The site of binding can be either NIm-IA of RV-B14 as shown for Fab17-IA [37] or NIm-II of RV-A2 [39]. Fab17-IA was shown to neutralise RV-B14 by reaching across the canyon blocking attachment, although one structural analysis demonstrated this Ab could undergo paratope conformational changes allowing canyon penetration [28]. Similar studies by Che et al. [25] demonstrated several Abs binding NIm-IA of RV-B14 make significant contact with the canyon that overlaps with the ICAM-1 binding site. Even though the canyon is largely inaccessible to Abs due to its narrow dimensions (known as the “canyon hypothesis” [40]), these studies identify that Abs can still interfere with entry receptor engagement without penetrating deep into the canyon. Therefore, Abs binding near the canyon can neutralise RVs by steric hindrance of entry receptor engagement which depends upon the angle of Ab attack and with bivalent binding to adjacent epitopes found across the 2-fold axis of the capsid. NIm-II binding nAb information is limited to few studies that indicate again that divalent binding across the 2-fold axis and stabilisation of the capsid structure is the most likely neutralising mechanism [41,42]. The angle of Fab attack has been shown to influence the valency of such nAbs that bind this region that still result in neutralisation [43]. Here, even though the adjacent epitopes are relatively close together, Ab 8F5 due to Fab arm flexibility, was shown to interact bivalently across the 2-fold axis over a shorter distance than originally anticipated. Such an angle of Fab attack reduces Ab occupancy but promotes neutralisation. Epitopes being too close together would only allow reduced valency monovalent binding and therefore not provide neutralisation capabilities due to the consequent loss of capsid structure stabilisation.

Similar mechanisms of neutralisation of the related picornaviruses HAV and enterovirus 71 (EV71) have been demonstrated. Potent neutralising Abs to EV71 that bind across the 2-fold axis and rigidify capsid structure have been found [44]. nAb R10 binds the 5-fold axis of HAV, mimicking the entry receptor and blocking attachment to cells [45] and nAb MA28-7 binds a similar region of the EV71 capsid, blocking entry receptor engagement and potentially aggregating virions due to monovalent binding [46]. Information about nAbs binding to the corresponding region of minor group RV serotypes is much more limited despite studies that demonstrate interference with soluble LDLR can neutralise minor group RV-A2 [47].

### 4.2. Ab Induction of RV Genome Release before Attachment to Cells

A key mechanism of RV entry into cells is the capsid conformational changes that lead to the process of uncoating where viral RNA is released into the cytosol [48]. Thus, an important mechanism of RV entry is the capsid structural changes that occur after attachment to cells and are critical for genome release via uncoating. This process should occur following endocytosis into endosomes, however a new neutralising mechanism of nAbs, upon binding to virions outside the cell, has been shown to cause the capsid conformational alterations resulting in RNA release prematurely.

Plevka et al., first demonstrated for EV71 that the nAb E18 bound the capsid and induced structural changes allowing for release of the RNA genome [49]. E18 appears to bind the EV71 capsid near the PUFF loop—analogous to NIm-II of the RV capsid. Similar observations have been revealed by Dong et al. for RV-B14 neutralisation [29]. This study reported that virus particles could be transformed into emptied particles upon nAb C5 binding to the NIm-III site. In contrast to poliovirus uncoating that occurs only under low ionic strength conditions, C5 Fab was shown to cause RV-B14 uncoating at physiological ionic strength. A finding suggesting that these antibodies could trigger viral genome release in vivo. C5 Fabs were shown to induce pore formation in the capsid near the 2-fold axis, a site that separates during initial uncoating. This is a different site to the major RV group uncoating and RNA release region, which is thought to be at the 5-fold axis raised plateau formed by VP1 residues, and is a unique site where non-neutralising Ab 2G2 binds to RV-A2 [50].

Similarly, for EV71 mAb A9 has shown to neutralise through capsid destabilisation and causing structural disruptions releasing RNA before the virus enters host cell [51]. This mAb also binds a region of the capsid spanning the 2-fold axis, inducing capsid separation. It certainly appears from multiple studies that Abs recognising a region spanning the picornavirus 2-fold axis are capable of multiple mechanisms of neutralisation. However, due to limited numbers of high-resolution structures of such RV-Ab complexes, the understanding of this mechanism of Ab-induced uncoating is restricted to specific examples and is still unclear if it is generalisable to many picornaviruses.

### 4.3. Prevention of Uncoating by Abs

Early studies investigating Ab neutralisation of picornaviruses have demonstrated the likelihood that stabilisation of the capsid structure and prevention of the structural changes that precede RNA release are another important mechanism [30]. Such a mechanism was termed “post absorption neutralisation” which suggests such Abs do not interfere with virus attachment to cells or could indeed function after attachment. Since many of the early studies would not have investigated this mechanism directly, examples of Abs neutralising RV in this way are somewhat limited.

Abs neutralising naked viruses by this mechanism may also act cooperatively and require fewer interactions than might be necessary for Abs that block attachment and require interaction at a specific capsid site, a concept proposed in 1984 to explain the stoichiometry of Ab interactions and synergistic effects of pairs of mAbs [22]. The observations for nAbs binding NIm-II of the RV capsid indicate that divalent binding across the 2-fold axis and stabilisation of the capsid structure is likely to be the neutralising mechanism. Thus, the well-studied mAb 8F5 binds bivalently across the 2-fold axis, does not obstruct the canyon and does not alter capsid structure [41,42], suggesting a potential post attachment mechanism. Furthermore, a similar nAb in terms of binding site, that does not bind divalently suggests that the angle of Fab attack influences nAb effectiveness and capsid stabilisation resulting in neutralisation [43]. For RV-B14, mAbs binding to NIm-IA that are classed as strong neutralisers are thought to stabilise the capsid from pH-induced inactivation which could indicate the mechanism of neutralisation [25]. This mechanism might therefore be equally interpreted as either prevention of uncoating or blocking attachment in the absence of more detailed investigations.

### 4.4. Aggregation of RV by Ab Binding

There are three critical factors that dictate virus aggregation by Abs and the contribution of this to loss of infectivity of RVs. Firstly, the physicochemical environment whereby aggregation could be replaced by a disruptive mechanism of neutralisation by lowering the ionic strength. Secondly, at least 10 times higher concentration of nAb over virus would be required for full aggregation and neutralisation. Thirdly, the preference of the nAb for monovalent binding and antigen recognition such that two identical Ab arms bind to two independent viral particles, creating a network of virus-Ab units as has been shown for poliovirus [27].

This mechanism has not been demonstrated conclusively for RVs but in the case of RV-B14, aggregation did not appear to have an important role in vitro as a NIm-IA binding Ab was a strong aggregator but only a weak neutraliser [25]. However, through opsonisation in vivo this mechanism of RV aggregation might assist protective immune responses. Earlier studies by Colonno et al. [26] using RV-B14 demonstrated that Abs targeting the other three neutralisation sites (NIm-IB, NIm-II, NIm-III) caused significant aggregation of virions which could be reversed by digesting virus-Ab complexes with papain, an enzyme that releases monovalent Fab units. Therefore, studies demonstrating this mechanism of neutralisation of RVs are restricted to few examples and are somewhat inconsistent, similarly to what is known for poliovirus. For example, 19 mouse mAbs to poliovirus type 1 displayed a variety of neutralising mechanisms and just one of these caused virus aggregation [30]. Taken together, this mechanism of RV and picornavirus neutralisation appears to be a relatively minor component, is determined by numerous properties of nAbs, and physiologically requires predominantly polyclonal responses.

### 4.5. Neutralisation Inside the Cell through TRIM21

It was thought that viruses escape Ab-mediated immunity once they infect cells. However, new findings suggest that certain viruses can enter cells attached to Abs, which can be recognised in the cytoplasm by a protein called TRIM21. This recognition induces intracellular pathways of destruction against these viruses. One key discovery is that TRIM21 is widely expressed in a large number of cells and tissues, thus, TRIM21 is well positioned to combine both innate and adaptive immunity to provide a second line of defence against intracellular pathogens such as viruses [52]. TRIM21 belongs to a large family of TRIM proteins that contain several critical domains such RING finger, zinc-binding fingers, B-box, and coiled-coil which function together for protein interactions and directing ubiquitin-protein ligase activity [53].

TRIM21 is a high affinity immunoglobin receptor, it is able to attach to IgG, IgA and IgM Abs in the cytosol through the Ab Fc interaction with the TRIM21 PRYSPY domain [54]. TRIM21 has been shown to provide antiviral protection against nonenveloped viruses, such as adenoviruses and RVs entering cells with attached specific Abs that are recognised by TRIM21 and subsequently targeted for degradation [31,55]. This mechanism of Ab neutralisation of RVs via TRIM 21 is known as antibody-dependent intracellular neutralisation or ADIN but the physiological significance remains unclear. ADIN has only been demonstrated for RV-A2 using polyclonal human serum [31]. Studies with mAbs have yet to be performed to fully determine this novel mechanism. Furthermore, information remains scarce regarding the optimal binding site of Abs that would not neutralise extracellularly and could therefore be potentially carried into cells to engage TRIM21. However, it is likely that such Abs would bind to non-neutralising epitopes of the RV capsid and would therefore be prevalent.

Finally, activation of TRIM21 by this mechanism induces the antiviral state within the cell and immune signalling through stimulation of activator protein 1 (AP-1), NF-kB and IRF3/IRF7 (interferon regulatory factors) in addition to ADIN [31]. These intracellular mechanisms mediated by Abs and TRIM21 provide another route to understanding of protective Abs for RVs. It is also possible that other TRIM proteins such as TRIM5α, TRIM19, TRIM22, and TRIM25 could mediate intracellular immunity to RVs. Additionally, TRIMs are upregulated by interferons (IFN), an early and important mediator of innate immunity to RVs [56].

### 4.6. Ab-Dependent Cellular Phagocytosis (ADCP)

The antiviral activity of Abs depends not only on its interactions with cognate antigens but also with immunoproteins and Fc receptor (FcR) expressing immune effector cells. The FcR-dependent Ab function combines the antiviral activity of innate effector cells with the diversity and specificity of the adaptive humoral immune response [34]. As such, the FcR-dependent function of Ab-dependent cellular phagocytosis (ADCP) contributes to the clearance of virally infected cells and of viral particles themselves. Ab antiviral effects in vivo are obviously more complex than virus neutralisation alone, especially when considering the additional components of the immune system that can assist with clearance. This role of non-neutralising Abs has come under increased scrutiny in recent years. ADCP activity has been shown to be an important mechanism for clearance of enveloped viruses such as influenza, HIV-1 and herpesviruses [57,58,59] but little information exists to support this role for clearance of picornaviruses.

ADCP effects of non-neutralising Abs are most likely to be more suited to enveloped viruses however for RVs, a recent study identified a mAb with cross serotype reactivity that demonstrated substantial ADCP activity [33]. In this study, both RV-A15 specific mAbs and cross reactive mAbs were identified. The specific mAbs but not the cross-serotype reactive Abs neutralised RV-A15 in vitro [33]. Further investigations revealed substantial ADCP activity for one cross-reactive mAb that bound shared regions of VP1. These data suggest that VP1-specific and cross-serotype binding RV Abs warrant renewed investigations of ADCP activity and may give insight into a novel mechanism of RV clearance by Abs. Thus, the role of non-neutralising Abs should not be ignored when considering protective responses to RV.

## 5. Summary and Implications for RV Vaccine Development

Protective Ab responses to RVs are established as a defined immunological correlate [60] but are notoriously serotype specific due to antigenic differences between the approximately 160 strains. Therefore, designing and implementing vaccines to protect against RVs has been extremely challenging for many years and has yet to result in an approved product [9]. This has led to the application of developing technologies such as antisense oligonucleotides or catalytic DNA enzymes (DNAzymes) as potential treatments for RVs [61].

Studies on RV Abs and protective immunity began in the 1960s and continued through the 1970s with several clinical trials investigating inactivated RV preparations as vaccine candidates (reviewed in [62]). After these approaches were abandoned, studies investigating Ab responses to RVs focused on the application of mouse monoclonal Abs and the identification of escape mutants which uncovered the existence of potentially four neutralising epitopes [16,17]. Whilst these studies were important and the use of mAbs brought the field further on, conclusions were difficult to generalise across all the RV serotypes and numerous conflicting structural studies, whilst helping to understand serotype specificity of mAbs, failed to establish a general consensus. We attempted to summarise these studies here and bring together the knowledge regarding Ab specificity for RVs and their mechanism of neutralisation. Ultimately, we wish to use this knowledge to design new approaches for generation of a successful RV vaccine.

We identified at least six potential mechanisms of Ab neutralisation or clearance for RVs as discussed. These range from steric hindrance of entry receptor engagement to Ab-induced structural changes or stabilisation of capsid structure that can influence uncoating mechanisms or adsorption to cells. Many of these observations were originally identified from analyses performed in vitro with purified Abs and could be expected to not fully reflect situations where Abs function in vivo—thus we included as a 6th mechanism the engagement of phagocytic cells through Ab Fc regions. Importantly, these Abs performing ADCP were probably dismissed previously as non-neutralising Abs but in light of new observations should be considered in the context of clearance of RV in vivo.

Therefore, how can this information be applied to vaccine development for RVs? Is it possible to induce Abs with a variety of neutralising properties with the vaccine platforms that are available? Can these immunogens be combined into a polyvalent vaccine formulation? Neutralisation of all RVs with a monovalent vaccine is very difficult and likely to fail due to the large numbers of antigenically distinct serotypes. However, several studies have shown that significant cross-serotype immunity and even neutralisation by Abs is possible and so enhanced immunogen design to allow immunisations that elicit the many nAbs responsible for the variety of neutralisation mechanisms outlined here could be a useful approach. The role of non-neutralising Abs should also not be discounted, particularly when these Abs tend to be cross-serotype reactive and could potentially induce ADCP of a large number of serotypes in vivo.

## Figures and Tables

**Figure 1 viruses-13-00360-f001:**
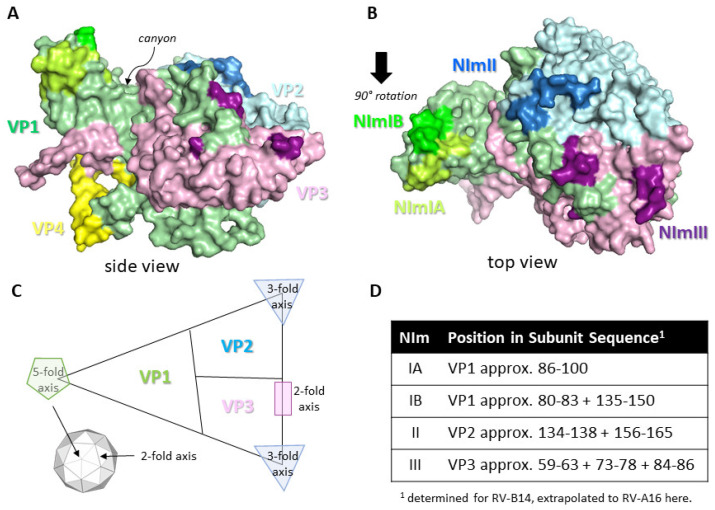
Protomer structure of RV-A16 and neutralising sites for Abs. Coordinates for RV-A16 protomer structure (1AYM) were downloaded from the protein data bank at www.rcsb.org and images created using PyMol molecular graphics system version 2.3.2 (Schrodinger LLC). (**A**,**B**) shows the protomer structure with VP1 (green), VP2 (light blue), VP3 (pink), and VP4 (yellow). Neutralising immunogenic (NIm) regions are highlighted light green (NIm-IA), lime green (NIm-IB), blue (NIm-II), and purple (NIm-III). The canyon region is identified in the side view shown in (**A**). (**C**) Schematic diagram of the RV capsid protomer showing approximate locations of the VP1, VP2 and VP3 subunits and the three axes of symmetry, including their position within the RV capsid icosahedron (smaller inset). (**D**) Location of NIm sites within each capsid protein subunit of RV-A16. Numbering refers to each individual capsid subunit amino acid sequence. NIm-IA is the only known linear sequence whereas the remaining NIms are discontinuous epitopes.

**Figure 2 viruses-13-00360-f002:**
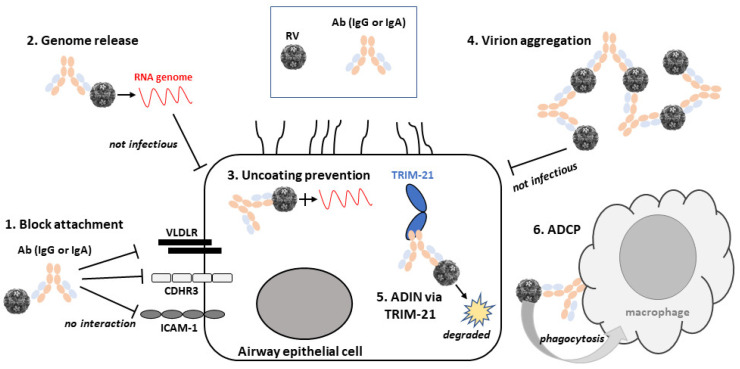
Mechanisms of rhinovirus (RV) neutralisation by Abs. A schematic diagram of an airway epithelial cell and the variety of neutralisation mechanisms of RV by Abs. Molecules are not drawn to scale. Abbreviations are as follows: RV, rhinovirus; Ab, antibody; VLDLR, very low-density lipoprotein receptor; CDHR3, cadherin-related protein 3; ICAM-1, intercellular adhesion molecule 1; ADIN, antibody-dependent intracellular neutralisation; TRIM-21, tripartite motif containing-21; ADCP, antibody-dependent cellular phagocytosis.

**Table 1 viruses-13-00360-t001:** Rhinovirus (RV) serotypes, strains and groupings.

	RV-A	RV-B	RV-C
Members	80 serotypes	32 serotypes	57 types ^3^
Features	Includes all minor ^1^ group RVs (10) and most of the major ^2^ group RVs (70)	All 32 are major group	1st discovery in 2007 and numbers continue to expand

^1^ those RV that use very low-density lipoprotein receptor (VLDLR) family as the entry receptor; ^2^ those RV that use intercellular adhesion molecule 1 (ICAM-1) as the entry receptor; ^3^ not designated serotypes yet as antibody neutralisation profiles not determined, entry receptor is cadherin-related protein 3 (CDHR3).

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
