# Peer review of "Mechanisms of Rhinovirus Neutralisation by Antibodies"

_viruses, 2021, doi:10.3390/v13030360_

Round 1

Reviewer 1 Report

Touabi et al. presentan overview of the possible mechanisms of neutralization of Rhinoviruses by antibodies.  This overview provides a good framework for categorizing the different ways antibodies can block Rhinovirus and includes mechanisms such as ADIN by TRIM21 which is only possible for proteinaceous viral particles like those for Rhinoviruses.  This is the type of information that could be useful in the design of a Rhinovirus vaccine although the high degree of diversity amongst Rhinoviruses makes this a challenge.  While the authors present useful information a number of things could be improved to make things clearer to the reader.

First, when structural information is available it would be useful to include some of that in a figure.  For example, the "canyon hypothesis" is not well described.  While figure 1 shows the 2-fold axis of the capsid it is not clear how that would fit into the capsid as a whole or how the angle of Fab attack would influence valency.  Are there examples of structures with higher and lower valenceis?  A figure could help clarify what is meant by this.

Second, when citing references it would be helpful to include more information about the reference.  For example, line 170 cites two mechanisms proposed for nAb inhibition of binding (reference 36 and 37).  It would be helpful to know R et al. proposed S based on T and X et al. proposed Y based on Z.  In general as a reader from outside the Rhinovirus field the more detail that is provided the more useful the review is to me. 

Some minor points.  The rotation arrow in figure 1 should show a rotation 90 degrees up from the position on the left and not from left to right.  line 40 "was" should be "were".  line 68 "of" should be "within".  

Author Response

We thank the reviewer for the careful consideration of the manuscript for for excellent suggestions. We have revised to satisfy the comments as follows:

The canyon hypothesis is not well described – we agree and have improved the text for this description between lines 201-207. In fact we have revised and added many parts of section 4.1 to make it more simple to follow for the reader.

Not clear how the 2-fold axis of the capsid fits into the whole capsid – we agree and have added an additional part to figure 1C demonstrating the 2-fold axis in the overall capsid structure and to the legend lines 117-118.

Are there examples of structures with higher and lower valencies?  A figure could help clarify what is meant by this. We have added test to clarify this on lines 212-217 and now feel that this aspect is explained much better. We are unable to construct a new figure with Ab-capsid structures and do not feel that it is required to understand the text now.

Second, when citing references it would be helpful to include more information about the reference.  For example, line 170 cites two mechanisms proposed for nAb inhibition of binding (reference 36 and 37).   We agree that this was confusingly written and have Added text to lines 193-195 to clarify this example and at subsequent locations to provide more details.

Minor points – we corrected the arrow orientation within figure 1 and uploaded a new figure. Text corrections have been applied to lines 41 and 70.

Reviewer 2 Report

With interest, I read the manuscript viruses-1117340. It is a very nice mini-review of the “Perspective” type.

I have several minor points:

  1. Considering that the classification of human RVs is based more and more on the genetics not serology, “serotypes” are now more and more frequently reacted with “types” (PMID: 30659817). Please, comment.
  2. Maybe I overlooked something but I cannot see the receptor for RV-C mentioned in your text. Please, provide (PMID: 30659817).
  3. Both figures are very nice but some their parts are too small. Could you possible make the figures bigger and horizontally oriented?
  4. The topic of the review is serology but some other developing approaches to combat RVs should also be exemplary mentioned, specifically those antisense-based (PMID: 30114391), optimally in chapter 5.
  5. All abbreviations used in the figures should be explained in the respective legends. It applies to both figures but especially to Figure 2.
  6. Likewise, “TRIM21” used in the Abstract should also be explained.

Author Response

We thanks the reviewer for the kind and careful review of the manuscript. We have revised the new version as follows:

  1. Considering that the classification of human RVs is based more and more on the genetics not serology, “serotypes” are now more and more frequently reacted with “types” (PMID: 30659817). Please, comment.

We agree about this – have revised the text to reflect this in lines 35-38.

  1. Maybe I overlooked something but I cannot see the receptor for RV-C mentioned in your text. Please, provide (PMID: 30659817).

It was not overlooked by the reviewer but not mentioned in the manuscript as no studies have demonstrated Ab blocking for RV-C. However we have now added details of the RV-C entry receptor to table 1 footnote and text (lines 92-93) plus new reference 15 added. We also added CDHR3 to Figure 2 diagram to highlight the potential R blocking mechanism.

  1. Both figures are very nice but some their parts are too small. Could you possible make the figures bigger and horizontally oriented?

This is an editorial setting that should be improved by the publisher. The images can be increased within the document. Both figures have been updated, added to the file and stretched as much as possible.

  1. The topic of the review is serology but some other developing approaches to combat RVs should also be exemplary mentioned, specifically those antisense-based (PMID: 30114391), optimally in chapter 5.

As requested we added a sentence to reflect this lines 362-363 and a new reference Potaczek et al that should be citation 61.

  1. All abbreviations used in the figures should be explained in the respective legends. It applies to both figures but especially to Figure 2.

Thank you for pointing out this omission. The figure legends now contain all abbreviations contained within the figures.

  1. Likewise, “TRIM21” used in the Abstract should also be explained .

Agreed, we replaced this with a general term in line 16 of the abstract.